



# Reviews and syntheses: Iron: A driver of nitrogen bioavailability in soils?

Imane Slimani[1,2], Xia-Zhu Barker[3], Patricia Lazicki[4], William Horwath[1]

[1]Department of Land, Air and Water Resources, University of California Davis, Davis, CA 95618, USA
[2]AgroBioSciences Program, Mohammed VI Polytechnic University (UM6P), Hay Moulay Rachid, Ben Guerir 43150, Morocco
[3]Department of Soil Science, University of Wisconsin-Madison. 1525 Observatory Drive. Madison, WI 53706-1299, USA
[4]Department of Biosystems Engineering and Soil science. University of Tennessee Knoxville, Tennessee 37996, USA

*Correspondence to*: Imane Slimani (islimani@ucdavis.edu)

**Abstract.** An adequate supply of bioavailable nitrogen (N) is critical to soil microbial communities and plants. Over the last decades, research efforts have rarely considered the importance of reactive iron (Fe) minerals in the processes that produce or consume bioavailable N in soils, compared to other factors such as soil texture, pH, and organic matter (OM). However, Fe is involved in both enzymatic and non-enzymatic reactions that influence the N cycle. More broadly, reactive Fe minerals restrict soil organic matter (SOM) cycling through sorption processes, but also promote SOM decomposition and denitrification in anoxic conditions. By synthesizing available research, we show that Fe plays diverse roles in N bioavailability. Fe affects N bioavailability directly by acting as a sorbent, catalyst, and electron transfer agent, or indirectly by promoting certain soil features, such as aggregate formation and stability, which affect N turnover processes. These roles can lead to different outcomes on N bioavailability, depending on environmental conditions such as soil redox shifts during wet-dry cycles. We provide examples of Fe-N interactions and discuss the possible underlying mechanisms, which can be abiotic or microbially meditated. We also discuss methodological constraints that hinder the development of mechanistic understanding of Fe in controlling N bioavailability and highlight the areas of needed research.

## 1 Introduction

Terrestrial ecosystem productivity is largely constrained by nitrogen (N) availability (Vitousek and Howarth, 1991). The largest pool of N in these ecosystems is found in soils which contains 133–140 Pg of total N globally within the first top 100 cm of soil (Batjes, 1996). A clear description of the factors controlling N bioavailability in soils is needed to design agricultural practices that meet crop demand and mitigate N loss to the environment. A large literature exists on the effects of soil texture, OM, mineral N inputs, pH, moisture, and microbial communities on N mineralization. However, geochemical factors, such as reactive Fe minerals, are rarely considered in N cycling, though they are often studied as vital components of carbon (C) cycling. Since C and N cycles are interconnected in soils (Feng et al., 2019; Gärdenäs et al., 2011), they should be regulated



by the same factors, including mineralogy type (Wade et al., 2018). Moreover, a series of observations in the literature highlight the involvement of Fe in N dynamics:

(a)   A large proportion of SOM is contained in associations with Fe minerals (Lalonde et al., 2012; Wagai and Mayer, 2007). The close proximity between the two components can trigger a myriad of interactions, including OM
stabilization.

(b)  Fe is a redox-active mineral that cycles between two redox states (Fe(II) reduced; Fe(III) oxidized). Fe(II)/Fe(III) redox transformations are tightly coupled with N cycling reactions (Kappler et al., 2021; Li et al., 2012a).

(c)  A myriad of interactions (Fig. 1) between Fe and N cycles have been observed in soils. These reactions, which can occur through both chemical or microbial pathways, include chemo-denitrification (Burger and Venterea, 2011) and
anaerobic ammonia oxidation coupled with Fe(III) reduction- Feammox (Wan et al., 2021). In addition, Fe is shown to affect rates of denitrification (Wang et al., 2016) and nitrification (Huang et al., 2016a) in experiments with both Fe addition and soil endogenous Fe (Han et al., 2018).

(d)  Increasing evidence shows that Fe represents a major control over N processes. For example, Fe (III) minerals and Fe complexed with SOM explained nitrous oxide ($N_2O$) emissions across a set of agricultural soils; more than any
other intrinsic soil property (Zhu et al., 2013). Similarly, Han et al. (2018) found that soil Fe regulates $N_2O$ emissions. By using structural equation modeling, Wade et al. (2018) found that Fe oxides strongly mediate N mineralization in agricultural soils.

(e)  Fe is involved in the enzymatic processes in the N cycle. For example, dissimilatory nitrate reductase, which catalyzes the first step in denitrification, contains Fe as a component of the internal electron transfer chain. Similarly, nitrite
oxidoreductase, which catalyzes ammonia oxidation to nitrite, contains Fe-rich cytochromes. Fe also regulates the expression of proteolytic genes responsible for protease production (Maunsell et al., 2006).

Therefore, the impacts of Fe on N cycling can be significant and should be considered. This review aims to understand the roles of Fe in controlling N bioavailability. To do so, we categorize the processes by which Fe affects OM dynamics into four different categories/roles. In the **sorbent role**, OM interacts with Fe(III) through adsorption, coprecipitation or surface coatings
(Eusterhues et al., 2005; Lalonde et al., 2012; Wagai and Mayer, 2007). These associations increase OM storage by decreasing its availability to extracellular enzymes and decomposition processes (Lalonde et al., 2012). In fact, the content of Fe minerals is a major predictor of soil sorptive capacity (Mayes et al., 2012). In the **structural role**, Fe minerals participate in the formation of soil aggregates (Zhang, X. et al., 2016) and increase soil structural stability (Barral et al., 1998; Xue et al., 2019). Aggregates can increase OM stability and retention in soils by protecting it from the decomposer community and their enzymes
(Kleber et al., 2021; Van Veen and Kuikman, 1990). Moreover, Fe(III) can facilitate the formation of large polymers of OM that promote its stability. Thirdly, Fe's **electron transfer role** depends on its oxidation state. Fe(III) serves as a sink of electrons, while  Fe(II) functions as a source of electrons. During anoxic periods, dissimilatory Fe(III) reduction can be coupled with the oxidation of OM, which accounts for significant amount of C loss under anoxic conditions (Dubinsky et al., 2010; Roden  and  Wetzel,  1996).  This  process  can  release  previously  adsorbed  or  coprecipitated  C,  thereby  increasing  its





susceptibility to degradation. Finally, Fe has a **catalysis role**, whereby Fe acts as a catalyst for the production of reactive oxygen species (ROS), which are potent oxidants of OM. This happens through Fenton reactions that are prevalent in various soils such as cultivated soils (Chen et al., 2020; Hall and Silver, 2013), arctic soils (Trusiak et al., 2018) and desert soils (Georgiou et al., 2015; Hall et al., 2012). These reactions are an overlooked but potentially important pathway for OM transformation in soils and sediments and N bioavailability (Kleber et al., 2021; Lipson et al., 2010; Merino et al., 2020;

Trusiak et al., 2018; Wang et al., 2017).

While these roles of Fe in controlling C cycling have been studied extensively, their effects on N bioavailability are not well explored. This review seeks to underpin these suggested relationships and provide mechanistic descriptions of how Fe controls N bioavailability in soils. This information is needed to construct reliable models with improved predictive power of N cycling in terrestrial ecosystems (Wade et al., 2018), and will offer new possibilities for land management.

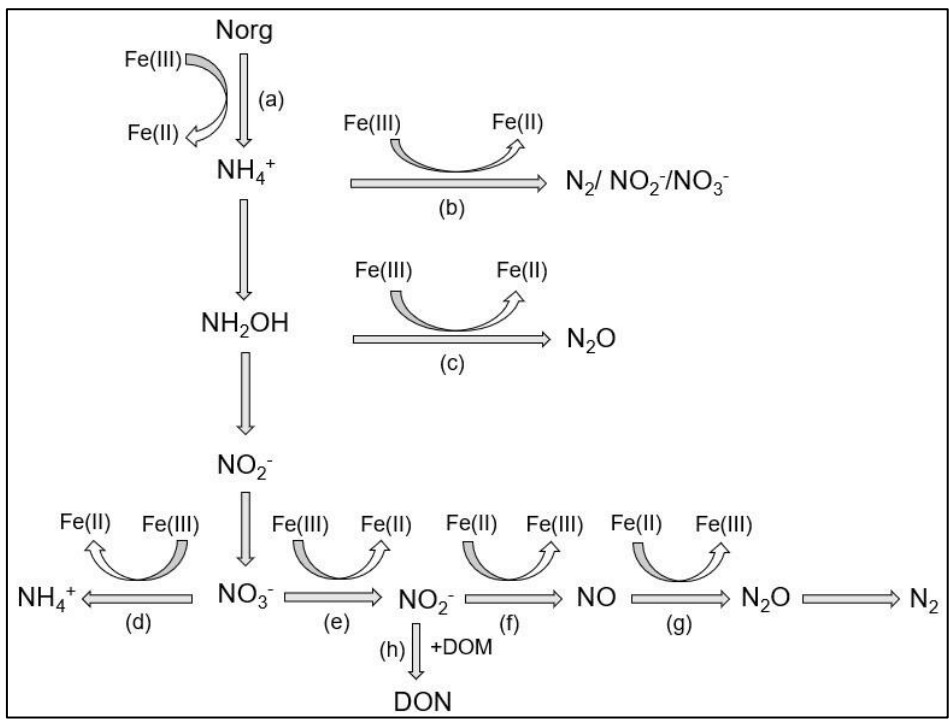


**Figure 1. Fe interacts with N cycles at various steps: (a) mineralization, (b) feammox, (c) N₂O production by Fe-mediated hydroxylamine oxidation, (d) anaerobic reduction of nitrate to ammonium, (e)+(f)+(g) Denitrification, (h) Fe-meditated abiotic formation of dissolved organic nitrogen (DON) by reaction of nitrite (NO₂⁻) with dissolved organic matter (DOM).**

## 2 Fundamental concepts to understand Fe-N interactions

The interactions of Fe and N depend on numerous characteristic properties of Fe and N compounds, which are briefly described below. The soil environment has the capacity to shape these interactions through changing these properties.





## 2.1 Iron

Iron oxides, hydroxides and oxyhydroxides (collectively referred to as iron oxides (Fe-oxides)); are widespread redox-active minerals in soil and sediments. Although Fe is relatively abundant in soils, the amount of its bioavailable fraction is low

(Colombo et al., 2013). In oxic soils, Fe mainly exists as poorly soluble Fe(III) oxides. In poorly drained soils experiencing periodic or transient anoxia, Fe(III) oxides undergo reductive dissolution, through both chemical and biotic pathways, which releases soluble Fe(II). This process is controlled by pH and redox potential (Eh) of soil, which control the distribution between Fe(III) and Fe(II) species.

Fe(III) oxides exist in a variety of polymorphs with unique physical and chemical properties controlling their reactivity

(Navrotsky et al., 2008). These minerals have variable surface charge generated by the protonation-deprotonation of surface hydroxyl groups, which provide a significant proportion of pH-dependent negative charge in soils (Sumner, 1963). The presence of pH-dependent charged groups influences the interactions with OM. When their relative point of zero charge (PZC) is below soil pH, Fe oxides exhibit positively charged surfaces and sorb negatively charged OM; however, if PZC is above soil pH, OM will be repulsed from the negatively charged Fe minerals.

The reactivity of an Fe mineral is also driven by surface topography, particle size and crystallinity. First, surface topography defects, which arise from deviations in the ideal composition and/or structure of minerals, play an important role in a variety of surface processes. Defect sizes and content are closely related to the strength and adsorption capacities of minerals (Li et al., 2015a), as they offer potential binding sites for organic molecules and promote the formation of stronger surface complexes (Petridis et al., 2014). Moreover, Fe(III) minerals with few defects are less vulnerable to reductive dissolution in favor of the

ones with more defects (Notini et al., 2019). Second, small particles possess high specific surface area (SSA), which enables high adsorption capacity. Finally, less crystalline and more disordered phases react readily because they typically have a much larger and more reactive SSA as well as higher solubility compare to crystalline phases (Schwertmann, 1991).

Additionally, Fe oxide reactivity is altered by the presence of OM coatings on mineral surfaces (Gao et al., 2018; Kleber et al., 2007; Poggenburg et al., 2018) and the type of coverage (monolayer vs. multilayer coverage). For instance, adsorbed organics

can inhibit the development of crystals (Boland et al., 2014; Henneberry et al., 2016), halt the reductive dissolution by surface passivation, reduce the amount of binding sites available for sorption (Kaiser and Zech, 2000a), or limit Fe(II) oxidation under oxic conditions (Daugherty et al., 2017).

## 2.2 Nitrogen

N in soils is made available to plants and microbes by N mineralization. i.e., the process by which organic N (ON) is

decomposed to mineral forms of N (MinN: ammonium ($NH_4^+$) and nitrate ($NO_3^-$)). ON predominates over MinN forms and can make up to 95 % of the bulk soil N in some surface soils (Knicker, 2011; Schulten and Schnitzer, 1997). ON exists in various chemical forms (Box 1), with a predominance of proteins and peptides. N in these compounds is generally not directly bioavailable due to molecular size constraints on microbial cell uptake (Schimel and Bennett, 2004). Depolymerization





reactions, carried out by the activity of extracellular enzymes (EE), transform these polymers into soluble, low molecular

weight (MW) organic monomers (e.g., short oligopeptides, amino acids). These reactions have long been considered the rate-limiting step in soil N cycling (Schimel and Bennett., 2004), however, recent research suggests that substrate availability can be as or more important (Noll et al., 2019). As both peptidase activity and protein sorption are affected by Fe minerals, Fe may drive gross amino acid (AA) production in soils. Once mineralized, N monomers are subject to three possible fates: first, they can be directly utilized by soil microorganisms or plants (Farrell et al., 2011; Geisseler et al., 2010). Second, they can be

transferred into associations with soil minerals (mineral-associated organic nitrogen; MAON) and further occluded within soil aggregates. Finally, they can be further mineralized to $NH_4^+$, due to the activities of extracellular and intracellular enzymes such as urease and AA oxidases (Geisseler et al., 2010). Recent research shows that the size of AAs available for mineralization is controlled by peptidase activity, but more so by substrate (protein) availability, both of which are affected by the interactions with Fe minerals. Therefore, Fe may drive gross AA production in soils (Noll et al., 2019).

**Box 1: Chemical forms of organic N in soils**

Soil ON exists predominantly as protein and peptides, and to a lesser extent as amino-sugars and nucleic acids (Kögel-Knabner, 2006). Proteins are intrinsically reactive towards soil minerals, due to a number of properties, including hydrophobicity, surface charge distribution, surface area, number and type of functional groups, conformation, and size. For instance, smaller proteins often have fewer available sites for sorption (Lützow et al., 2006). Protein decomposition is often equated with complete depolymerization to AAs, however, small peptides and AAs can be produced in equal amounts (Warren and Taranto, 2010). Relative to AAs, peptides are preferentially and rapidly utilized by microbes as sources of C and N ( Geisseler et al., 2010; Farrell et al., 2011; Farrell et al., 2013; Hill et al., 2012) . Amino-sugars, which account for 5–8% of ON (Amelung et al., 1996), comprise chitin and peptidoglycan (PGN) and other components of microbial cell walls. The important contribution of amino- sugars to bioavailable N in soils is debated (Martin and Haider, 1979; Kögel-Knabner, 2002; Roberts et al., 2007; Strickland and Rousk, 2010; Roberts and Jones, 2012; Hu et al., 2018). Finally, nucleic acids are generally decomposed by nucleases and yield individual nucleotides in soils. The chemical composition of these compounds may affect their decomposition dynamics. For instance, adenosine monophosphate is degraded faster than cytidine monophosphate (Therkildsen et al., 1996).

## 3 Sorbent role of Fe in controlling N bioavailability

### 3.1 Does extracellular enzymes sorption to Fe oxides affect their participation in N mineralization?

Soil microbes produce a variety of extracellular enzymes (EE) to acquire N, and increased N-acquiring enzyme activities correlate positively with N mineralization. These enzymes can be substrate-specific (e.g., proteases and aminopeptidases), or

non-specific oxidative enzymes (e.g., laccase and peroxidase) (Caldwell, 2005; Sinsabaugh et al., 2009; Hassan et al., 2013), which are generally associated with C cycle, though their importance for N mineralization has also been demonstrated (Kieloaho et al., 2016; Zhu et al., 2014). Many of these enzymes become adsorbed to Fe minerals when released in soil. Such





immobilization often lowers enzyme activities, increases their resilience to proteolysis (Sarkar and Burns, 1984; Rani et al., 2000; Tietjen and Wetzel, 2003; Kelleher et al., 2004), and allows for greater residence time in soils and more persistent

activity (Yan et al., 2010; Schimel et al., 2017). However, opposing outcomes on enzyme activity have been reported (Quiquampoix and Ratcliffe, 1992; Quiquampoix et al., 1995; Servagent-Noinville et al., 2000). For instance, Fe adsorption reduced the activity of urease (Gianfreda et al., 1995; Bayan and Eivazi, 1999; Li et al., 2020), but increased the activity of N-acetyl-glucosaminidase (NAG) (Allison, 2006; Olagoke et al., 2020). These contradicting effects can have multiple explanations. First, enzyme active sites can become occluded, which limits the diffusion of N substrates towards the binding

sites and lowers N decomposition as a consequence. Site occlusion is due to either conformational changes in the enzyme structure (Datta et al., 2017), Fe-induced aggregation (Olagoke et al., 2020) or unfavorable attachment orientation on mineral surfaces (Baron et al., 1999; Yang et al., 2019). Second, Fe oxides can inhibit the activity of EE by constraining N substrate availability. Along a 120-kyr-old chronosequence, Turner et al., (2014) found that Fe oxides inhibited the activities of urease and proteases more strongly than aminopeptidases, possibly due to the preferential adsorption of urea and proteins over

peptides (Turner et al., 2014). Third, enzyme activity is likely affected by soil mineral content. Olagoke et al., (2020) observed that soil with low mineral content offers a limited availability of adsorption sites, allowing less and weak bonding of enzymes with minerals with minimal impact on enzyme active site. Therefore, enzymes in mineral-poor soils may have high and more persistent activities than those in mineral-rich soils. In this case, the presence of functional and active EEs may allow microbes to invest in biomass production instead of enzyme production, which results in improved microbial C and N use efficiencies

in mineral-poor soils, as hypothesized by (Olagoke et al., 2020). Other soil properties such as pH control enzyme sorption by affecting surface affinity and related binding strength and enzyme conformation (Quiquampoix et al., 1993). Finally, a new mechanism has been proposed recently by Chacon et al., (2019), who observed (experimentally) that goethite can induce the abiotic fragmentation of proteins and subsequent loss of activity (Chacon et al., 2019). The occurrence of this mechanism in soil and implications for enzyme activity and N bioavailability awaits further investigation and validation. Beyond adsorption,

enzyme activity is affected by soil redox conditions. For instance, waterlogging treatments decreased the activity of urease (Pulford and Tabatabai, 1988; Gu et al., 2019) , whereas the activity of amidase was not affected (Pulford and Tabatabai, 1988). These effects were attributed to the production of reduced metals under waterlogged conditions, which may serve as inhibitors or activators of enzymes (Pulford and Tabatabai, 1988). Specifically, Fe(II) was shown to stimulate the activities of oxidative enzymes under anaerobic conditions (Van Bodegom et al., 2005; Sinsabaugh, 2010), but strongly inhibit the activity

of urease (Gotoh and Patrick Jr, 1974; Tabatabai, 1977). To conclude, Fe affects N-acquiring enzymes differently depending on the modalities of their interaction, enzyme and substrate identity, and soil properties and conditions. The direction and the magnitude of this effect may create distinct patterns of N bioavailability and enzyme activities across soils (Turner et al., 2014).

**3.2 Does the sorption of N substrates to Fe oxides affect their bioavailability?**

Many studies have demonstrated that poorly crystalline Fe minerals, such as ferrihydrite, control the sorption of N compounds

in soils (Kaiser and Zech, 2000b; Dümig et al., 2012; Keiluweit et al., 2012a; Dippold et al., 2014). Indeed, Fe minerals interact





with a wide range of N-containing moieties via adsorption or coprecipitation processes; the latter process incorporates N into organo-mineral associations (MAOM), which are essential for OM stabilization (Leinweber and Schulten, 2000; Keiluweit et al., 2012b; Swenson et al., 2015; Heckman et al., 2018; Zhao et al., 2020). During these processes, Fe can form strong chemical bonds with N-containing moieties; for instance, goethite forms stronger bond with ammonia ($NH_3$) than with carboxylate,

phosphate, or methyl groups (Newcomb et al., 2017). The bond strength between N and mineral surfaces varies considerably across different environments due to differences in the nature of binding mechanisms, mineral and N properties, soil properties such as pH and ion strength, and the presence of antecedent SOM on mineral surfaces (Lützow et al., 2006). However, protein may adsorb irreversibly to mineral surfaces over a wide range of solution pH and resist desorption (Hlady and Buijs, 1996; Yu et al., 2013); the latter mechanism is perceived to be a necessary step for EE to proceed with N mineralization. Similarly,

nucleic acid molecules persist for a long time on clay minerals (Yu et al., 2013) and are shielded from degradation.

Advances in spectroscopic techniques have generated new conceptual models of organo-mineral associations, such as "the zonal structure model of organo-mineral associations", which postulates that organic compounds self-organize on mineral particle surfaces (Kleber et al., 2007). In this model, amphiphilic SOM compounds with N-bearing and oxidized functional groups directly interact with mineral surfaces to form "the contact zone", whereas hydrophobic groups face outwards creating

a region of high hydrophobicity, "the hydrophobic zone". Additional organic molecules attach to this zone, forming an outer layer termed "the kinetic zone". Multiple recent observations support this model, including (1) the preferential enrichment of N-containing moieties on Fe mineral surfaces (Kopittke et al., 2018; Possinger et al., 2020), (2) the preferential adsorption of N compounds over other organic compound classes on Fe mineral surfaces (Gao et al., 2017) and (3) the partial sorption of some organic compounds, including AAs, to Fe minerals (Amelung et al., 2002; Dippold et al., 2014). This model has

implications for N bioavailability, because, in contrast to the contact zone, the weakly sorbed N in the kinetic zone likely exchange with soil solution and is more available. Recent research on the chemical composition of C and N at the organo-organic and organo-mineral interfaces of the model found that alkyl C and less N occurred at the former, whereas oxidized C and more N occurred at the latter (Possinger et al., 2020). The authors of this study hypothesized that the processes stabilizing C and N at these interfaces are different, considering that the association between SOM rich in O/N-alkyl C and Fe oxides

explained the stabilization of O/N-alkyl C in soils (Schöning et al., 2005). In addition to protecting a fraction of bioavailable N, Vogel et al., (2014, 2015) found that sorption can retard the movement of N in soils, thereby increasing N retention by decreasing its accessibility to degradation mechanisms ( Vogel et al., 2014; Vogel et al., 2015). More insight is needed to advance the understanding of N bioavailability from organo-mineral associations.

## 3.3 N sorption is counteracted by several destabilization mechanisms

The release of N from Fe-organic associations, or desorption, occurs due to several destabilization mechanisms, including surface displacement by competitive sorption, oxidative and reductive dissolution of Fe minerals (Kleber et al., 2015) and local disequilibrium in soil chemistry. Once released, SON may become accessible and vulnerable to microbial degradation or





diffusion into microbial cells. The following is a discussion of the different destabilization mechanisms of Fe-organic associations in soils and factors influencing them:

(a)   N desorption by oxidation and reductive dissolution of Fe minerals

The dissolution of Fe minerals, as a result of changes in soil pH or redox conditions, decreases their sorption capacity and compromises the stability of sorption complexes. When mineral dissolution occurs, Fe and OM enter the soil solution. For instance, chemical reduction of Fe(III) by sodium dithionate was shown to release C and N substances compared with no reduction (Bird et al., 2002). However, short-range order (SRO) Fe oxides can resist both chemical and microbial reduction,

due to coprecipitation with aluminosilicates or physical protection within microaggregates (Henneberry et al., 2012; Shimizu et al., 2013; Eusterhues et al., 2014; Filimonova et al., 2016; Suda and Makino, 2016; Coward et al., 2018; Tamrat et al., 2019). The extent of OM mobilized from mineral reduction remains unpredictable due to knowledge gaps related to their resistance mechanisms and their controls in soils. The oxidation of Fe(II) can also release OM by solubilizing Fe-organo associations via decreasing pH or by generating hydroxyl radicals through Fenton chemistry, which oxidize OM abiotically. Redox fluctuations

can also affect OM cycling by changing mineral properties; for instance, such fluctuations can induce the transformation of amorphous Fe minerals into more crystalline forms, which can decrease OM stability and increase its turnover rates. However, mineral crystallinity was found to be positively correlated with SOM turnover rates (Hall et al., 2018) and was not associated with C release from Fe associations (Chen et al., 2020). These observations can be explained by the zonal structure of organo-Fe associations, in which OM in the kinetic zone can be lost, and the contact zone organics remain protected.

(b)   N desorption by local disequilibrium in soil chemistry

OM in soils can be desorbed from mineral surfaces due to the establishment of local disequilibrium conditions. Such conditions result from depletion of DOM in the soil solution, due to microbial uptake, for example, promote the release of OM from MAOM until DOM concentrations in the soil solution are in equilibrium with sorbed OM. This process is likely affected by the strength of bonds between N substrates and Fe minerals; in fact, interaction forces vary considerably: strong interactions

are favored by polyvalent cation bridges and ligand exchange whereas weak interactions occur by hydrogen bonds or van der Waals (Kleber et al., 2015). While the relationship between particular binding mechanism and N desorption from minerals has not yet been established in real soil conditions, multiple studies in model systems demonstrated that OM bound by ligand exchange was more resistant to desorption than other mechanisms (Wang and Lee, 1993; Gu et al., 1994; Gu et al., 1995; Mikutta et al., 2007). Therefore, it will be likely less affected by the dynamic equilibria principle and less N will be made

available (Kleber et al., 2015).

(c)   N desorption by surface displacement via competitive sorption

N associated with Fe can be displaced by the input of highly sorptive organic compounds. For instance, Scott and Rothstein (2014) observed that weakly bound, N-rich hydrophilic compounds were easily displaced by stronger binding compounds (e.g. hydrophobic compounds), leading to the downward migration of N to subsurface and mineral horizons.

(d)   Is desorption of N from organo-mineral associations a prerequisite to N mineralization?





As mentioned earlier, desorption of protein from mineral surfaces is often perceived to be the primary pathway by which N substrates become accessible to microbial degradation (Schimel and Bennett, 2004). However, protein adsorption to Fe minerals is an irreversible process (Rabe et al., 2011), which restricts proteolytic activity. Recently, the direct proteolysis of protein at the mineral surface was investigated, as ferrihydrite- and goethite-adsorbed protein was found to be degraded without

prior desorption (Tian et al., 2020). Substrate-enzyme complexes were formed directly at the surface of minerals. Together with the zonal structure of organo-mineral associations, this finding challenges the long-standing assumption that Fe minerals impair protein bioavailability through acting as a sorbent. The reader is referred to Keiluweit and Kuyper (2020) for a more expanded discussion of this mechanism (Keiluweit and Kuyper, 2020).

## 4 Structural role of Fe in controlling N bioavailability

### 4.1 Does structural Fe in clay minerals affect N bioavailability?

The majority of clay minerals contain Fe and account for 30-50 % of total Fe in soils and sediments (Favre et al., 2006; Stucki, 2013). Fe can be located in both the octahedral and tetrahedral sheets of 1:1 and 2:1 clay mineral or exist as coating on their surfaces (Stucki, 2013). N bioavailability can be affected by the redox cycling of this structural Fe in clays. For instance, the reduction of structural Fe(III) allows the abiotic fixation of $NH_4^+$ (Zhang and Scherer, 2000; Deroo et al., 2021) through

increasing negative charge and cation exchange capacity of clays (Pentráková et al., 2013). Further, the reductive dissolution of coated Fe on clay minerals promotes $NH_4^+$ diffusion into or out of clay interlayers (Zhang and Scherer, 2000). After de-fixation, the fixed $NH_4^+$ pool can serve as a source of bioavailable N (Deroo et al., 2021). In contrast to Fe(III) reduction, structural Fe(II) oxidation has not received much attention despite its possible involvement in processes that cause the loss of bioavailable N. For instance, Zhao et al., (2013) found that the oxidation of structural Fe(II) in nontronite causes the loss of

$NO_3^-$ as dinitrogen ($N_2$) (Zhao et al., 2013). The potential importance of such processes in N bioavailability should be considered, especially in highly weathered soils with high clay content.

### 4.2 Fe, soil aggregates and N bioavailability

Few studies have explored relationships between Fe, soil aggregates, and turnover of N in soils, despite multiple indications of their interconnection. First, Fe oxides are one of the most important constituents of soil microaggregates (Peng et al., 2015),

serving as nuclei for their formation and meditating their stability (Barral et al., 1998; Pronk et al., 2012; Wei et al., 2016), acting as a cementing agent (Colombo and Torrent, 1991; Krause et al., 2020) and binding OM (Giovannini and Sequi, 1976; Totsche et al., 2017). Second, Fe oxides preferentially adsorb N-containing moieties. The observations that C:N ratio of sorbed organics decrease with decreasing particle size (Aufdenkampe et al., 2001) and increasing particle density (Sollins et al., 2006), suggest that N is an important component of microaggregate-SOM (Golchin et al., 1994). Indeed, using density fractionation,

Wagai et al., (2020) observed joint accumulation of OM with low C:N ratio and pedogenic Fe and Al oxides in the meso-density fractions (1.8–2.4 $g\ cm^{-3}$) of five soil orders collected from different climate zones. Moreover, Rodionov et al., (2001)





observed high concentrations of amino sugars in microaggregates (Rodionov et al., 2001). These observations have implications for N bioavailability, given the facts that mineral-associated OM compounds located in stable aggregates has lower availability to microbes than those located on more accessible surfaces. Microaggregate-N is relatively more persistent

than macroaggregate-N because microaggregates' turnover is relatively slow, which provides longer-term stabilization of OM (Cambardella and Elliott, 1993; Six et al., 2002). Krause et al., (2020) demonstrated that colloidal sized Fe promotes the formation of smaller-sized microaggregates (<20 µm). In addition, readily mineralizable N levels correlate positively with increased aggregate size in soils (Mendes et al., 1999), suggesting that Fe mediated micro-aggregation may slow down or suppress N mineralization. We hypothesize that there is another pathway by which Fe-promoted aggregation may decrease N

mineralization. Aggregates of different sizes influence microbial community composition differently and therefore the activities of N mineralization enzymes (Muruganandam et al., 2009). Therefore, it will be useful to examine the distribution and the activities of these enzymes among soil aggregate size classes along a gradient of increased Fe mineral content in soils. The relative importance of Fe in aggregate stability depends on several properties, such as Fe mineral and SOM content, mineral identity and degree of crystallinity, and soil redox conditions, which are expected to affect N bioavailability. In

particular, Fe promotes the formation and stability of aggregates in soils with low OM and high Fe content (Barral et al., 1998; Wu et al., 2016). Duiker et al., (2003) showed that poorly crystalline Fe minerals are more important than crystalline minerals for aggregate stabilization (Duiker et al., 2003). Partial or complete removal of mineral-forming components, for example due to Fe reduction, can initiate aggregate turnover and destabilization (Michalet, 1993; Cornell and Schwertmann, 2003) which will eventually expose associated OM to microbial degradation (Lützow et al., 2006). Indeed, Cambardella and

Elliott (1993) found that the loss of aggregates caused organic carbon (OC) and ON loss from SOM (Cambardella and Elliott, 1993). Silva et al., (2015) reported that applying Fe-rich biosolids

in a tropical soil chronosequence induced rapid formation of microaggregates and significantly increased SOC (Silva et al., 2015). Similarly, Bugeja and Castellano (2018) observed positive correlation between ammonium oxalate-extractable Fe (AmOx-F), C and N in microaggregate, indicating that Fe and microaggregate stabilization are interconnected (Bugeja and

Castellano, 2018).

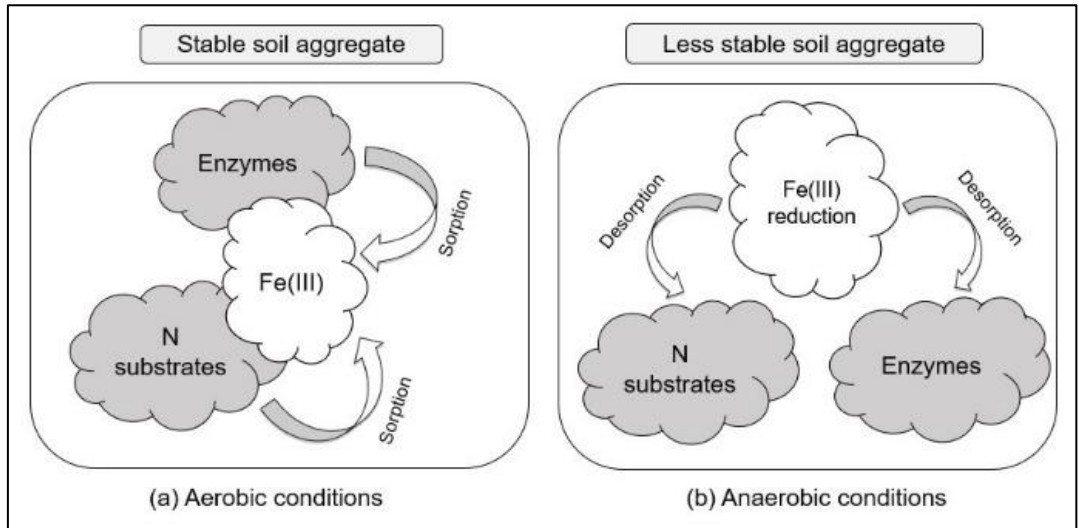

**Figure 2. (a) Fe minerals sorb both enzymes and N substrates and promote stability of microaggregates, which offers protection to N from degradation in soil under oxic conditions. (b) Fe reduction releases N substrates and may lead to aggregate destabilization in soil under anoxic conditions.**

### 4.3 Does Fe-induced ON polymerization increase the recalcitrance of N?

Little is known about Fe (mineral)-induced OM polymerization in soils. Some evidence exist that Fe oxides induce both C and N polymerization of SOM (Piccolo et al., 2011; Li, C. et al., 2012; Johnson et al., 2015; Zou et al., 2020) . In a long-term organic fertilization experiment, Yu et al. (2020) proposed that the Fe-catalyzed formation of reactive oxygen species (ROS) allows C monomers to recombine into large, recalcitrant C biopolymers through the formation of intramolecular bonds. A similar process was observed by Piccolo et al. (2011). Similarly, hydrohematite, maghemite, lepidocrocite and hematite can induce the oxidative polymerization of hydroquinone, with rates depending on the type of minerals (Huang, 1990). Synthetic ferrihydrite and goethite were demonstrated to induce peptide bond formation between aspartate chains (Matrajt and Blanot, 2004), as well as the abiotic formation of AAs from simple organics such as pyruvate and glyoxylate (Barge et al., 2019). The environmental conditions in these experiments were similar to those occurring in natural systems such as in Fe-containing sediments (Barge et al., 2019). More studies of abiotic polymerization by minerals must be envisaged given that sorption is a ubiquitous and naturally occurring phenomenon in soils.

### 5 Catalytic role of Fe in controlling N bioavailability

Emerging research has revealed that ROS derived from Fe-catalyzed Fenton reactions (Box 2) are implicated in N mineralization. These reactions may involve abiotic or coupled biotic-abiotic processes causing N to mineralize, as explained below. In desert soils, the reaction of light with hematite generates ROS, which can oxidize AAs to nitrous oxide ($N_2O$) (Georgiou et al., 2015) and N oxide gases (Hall et al., 2012). Compared to soil containing water, desert soils accumulate





photogenerated superoxides and peroxidases via complexation of $O_2^{\cdot-}$ with surface transition metal oxides. When these soils are wetted, the accumulated ROS are subjected to dismutation and hydrolysis leading to the generation of $HO^\cdot$ and subsequent OM oxidation. While this mechanism is strictly abiotic, soil microorganisms in diverse ecosystems were found to use Fe-

generated $HO^\cdot$ to acquire organic C and N (Diaz et al., 2013; Shah et al., 2016; Zhang, J. et al., 2016; Op De Beeck et al., 2018). For instance, a boreal forest fungus (*Paxillus involutus*) may use radical oxidation to stimulate N mineralization in various ways (Op De Beeck et al., 2018): (1) to liberate $NH_4^+$ from amine groups of proteins, peptides, and amino acids according to mechanisms reviewed in Stadtman and Levine (2003), (2) to facilitate the accessibility of protein-N in SOM complexes to proteolytic degradation and (3) to enhance protein vulnerability to proteolysis and increase the activity of

proteolytic enzymes (Zhang, J. et al., 2016).

Despite their involvement in N liberation, ROS may promote the formation of stable and protective Fe-associated OM complexes. In a long-term fertilization experiment conducted by Yu et al., (2020), Fe mobilized by Fenton reactions formed new short-range order (SRO) Fe minerals, which promoted C and N storage. Moreover, ROS generated from catalytic reactions involving Fe can also cause enzyme oxidation and subsequent loss of activity (Huang et al., 2013).

**Box 2: Fe-catalyzed Fenton reactions**

Most Fe minerals, such as ferrihydrite, goethite, hematite, magnetite, and pyrite, can catalyze Fenton-like reactions (Kwan and Voelker, 2003; Garrido-Ramírez et al., 2010). Fe-catalyzed Fenton reactions are mainly driven by fluctuating redox conditions (Xu , J.et al., 2013), oxygenation of $Fe^{II}$- bearing minerals (Tong et al., 2016) and photochemistry (Georgiou et al., 2015). Despite having a short lifetime in soil (Apel and Hirt, 2004), ROS, such as $HO^\cdot$ ($E^o$ = 2.8 V), are non-selective and strong oxidants of OM (Gligorovski et al., 2015).

Photoreduction of Fe(III)-ligand (L) complexes : Fe(III)-L + hv -> Fe(II) + L*

Reactions of Fe mediated ROS generation:

$Fe(II) + H_2O_2 \rightarrow Fe(III) + OH^- + HO^\cdot$

$Fe(II) + H^+ + HO_2^\cdot \rightarrow Fe(III) + H_2O_2$

$Fe(II) + O_2 + H^+ \rightarrow Fe(III) + HO_2^\cdot$

**6 Electron transfer role of Fe in N bioavailability**

Electron transfer to Fe(III) oxides, both biotically or abiotically, is a critical step in many processes favoring the gain or the loss of N from soils and sediments (Ding et al., 2014; Sahrawat, 2004). The ability of Fe(III) minerals to accept electrons, or their 'reducibility', varies greatly with crystallinity, particle size, solution pH, ambient Fe(II) concentration, the presence of

adsorbates and aggregation level (Roden, 2004; Roden, 2006). Here, we explore relationships between mineral reducibility and anaerobic $NH_4^+$ oxidation associated with Fe reduction (Feammox) and anaerobic OM oxidation to illustrate two examples of N processes that are involved in bioavailable N production and loss. Starting with Feammox, this process occurs mostly in acidic soils and has been estimated to metabolize 7.8−61 kg $NH_4^+$ /ha/year in paddy soils, accounting for about 3.9 %−31 %





of N fertilizer loss (Ding et al., 2014). The terminal products of this process are either $N_2$, $NO_2^-$ or $NO_3^-$ with $N_2$ as the dominant

product (Yang et al., 2012). Feammox rates are strongly positively correlated with the concentrations of microbially reducible Fe(III) ( Ding et al., 2014; Li et al., 2015b; Ding et al., 2019; Ding et al., 2020). Moreover, Fe(III) enhances the activity, distribution and diversity of microbial communities involved in Feammox (Huang, S. et al., 2016; Ding et al., 2017). A series of incubation studies investigated the effects of different Fe sources on Feammox, and the results demonstrated that only ferrihydrite and goethite, not ferric chloride, lepidocrocite, hematite, or magnetite, served as electron acceptors for Feammox

(Huang and Jaffé, 2015; Huang and Jaffé, 2018). These observations can be explained by a possible accumulation of free Fe(II), which halted Feammox, or due to the limited ability of Fe-reducers in reducing certain minerals (Huang et al., 2014). It is notable that chelates (Park et al., 2009) and electron shuttles (Zhou et al., 2016) can facilitate electron transfer to Fe(III) minerals (Fig. 3), which enhances their reduction rates and related N processes. For instance, the addition of electron shuttles increased potential N loss by Feammox by 17–340% compared to no addition (Zhou et al., 2016). Similar to Feammox, $NH_4^+$

production rates in submerged soils and sediments were found to be strongly correlated with reducible Fe(II) production rates (Sahrawat and Narteh, 2001; Sahrawat, 2004).

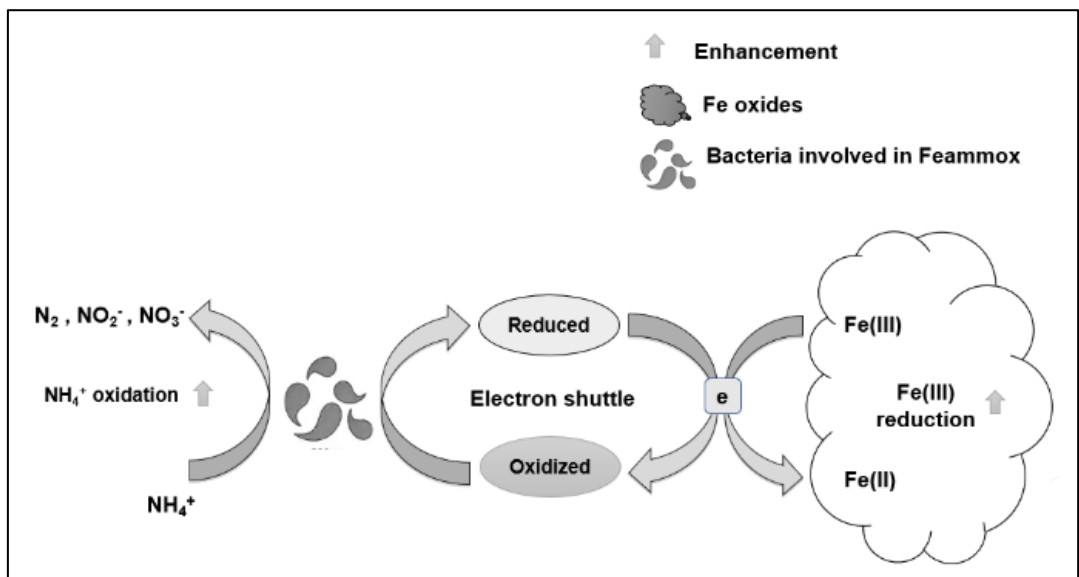

**Figure 3. Electron shuttles enhance Fe reduction and NH4+ oxidation (Feammox) rates**

The electron-donating capacity of Fe minerals is also involved in N bioavailability. In fact, many Fe(II) species, including

soluble Fe(II)- and Fe(III)-bearing minerals such as siderite and magnetite, can act as electron donors (Benz et al., 1998; Chaudhuri et al., 2001) for $NO_3^-$ reduction coupled with Fe oxidation, which promotes the loss of $NO_3^-$ as gases. For denitrification, it was found that $N_2O$ emissions from flooded soils with contrasting Fe(II) levels were regulated by Fe(II) electron donating capacity: the electrons donated reached 16.2% and 32.9% in soils with low and high Fe(II) content, respectively. Soil with high Fe(II) content emitted less $N_2O$ and more $N_2$, suggesting an improved denitrification efficiency

due to an electron flow which exceeded the demand for $N_2O$ production (Wang et al., 2016).



## 7 Involvement of Fe in soil phenomena that affect N bioavailability

### 7.1 Priming

Priming occurs when new input of labile C influences (positive or negative) the decomposition of native SOM (Kuzyakov et al., 2000). Several mechanisms have been proposed to explain this effect, including a shift in microbial communities (Fontaine
et al., 2003), microbial N mining (Craine et al., 2007) and microbial activation (Drake et al., 2013). However, investigations of the patterns and drivers of priming across both local and broad geographical scales indicate that SOM stabilization mechanisms, including associations with Fe oxides, regulate priming and explain most of its variation (Chen et al., 2019; Jeewani et al., 2021a). In fact, positive priming, which occurs when new inputs increase SOM mineralization, is negatively related to MAOM concentration due to Fe constraining the accessibility of sorbed organics to microbial degradation (Bruun
et al., 2010; Porras et al., 2018). Thus, the disruption of Fe-organic associations can lead to positive priming by liberating sorbed C and N compounds and making them more accessible. In the rhizosphere, this process takes place chemically when plant exudates strip Fe from Fe-organic associations by surface complexation, displacement of sorbed organics into soil solution (Keiluweit et al., 2015) and reductive dissolution of Fe (Zinder et al., 1986; Ding et al., 2021). Biotically, root exudates can activate microbes by providing C and energy, leading to increased production of N-acquiring enzymes and subsequent N
mineralization (Yuan et al., 2018; Jilling et al., 2018; Jiang et al., 2021; Jilling et al., 2021).

The magnitude of priming depends on the extent to which these aforementioned destabilization pathways affect Fe-organic associations. For instance, Li, H. et al., (2021) showed that MAOM on ferrihydrite is susceptible to both abiotic and biotic pathways, whereas MAOM on goethite is more susceptible to abiotic pathways (Li, H. et al., 2021). Therefore, the ability of microbes and plant communities to secrete specific exudates capable of triggering specific destabilization pathways of the
dominant mineral in their environment will affect how much N can be made available from mineral associations (Jilling et al., 2018; Li, H. et al., 2021).

### 7.1.1 Fe-mediated priming in soils under reducing conditions

Recently, Fe-mediated priming in soils under reducing conditions has received growing interest. Dunham-Cheatham (2020) found that glucose application to a soil under anoxic-oxic transition induced a novel type of priming by facilitating the reductive
dissolution of Fe$^{III}$-C associations under anoxic conditions followed by a dramatic increase of OC mineralization when oxic conditions were restored (Dunham-Cheatham et al., 2020). Li, H. et al., (2021) found that the roles of Fe in anaerobic OM mineralization can be shifted by microbial biomass C (MBC). In soil with low MBC, both ferrihydrite and goethite protected the added acetate from decomposition through sorption processes. In soil with high MBC, however, goethite acted as an electron acceptor and increased acetate decomposition, whereas ferrihydrite predominantly adsorbed the added substrate.
Priming decreased in both low and high MBC soils, but more in low MBC soil (Li, H. et al., 2021). Lecomte et al., (2018) demonstrated that Fe(III)-reducing microorganisms have a competitive advantage of colonizing plant roots in the rhizosphere due to their capacity of providing Fe(II) for plant nutrition in exchange for C-rich exudates and performing denitrification

(Lecomte et al., 2018). These exudates are probably used as a C source in the denitrification process or to destabilize Fe-organic associations and release sorbed C and N (Dunham-Cheatham et al., 2020). More research into Fe-mediated priming in strictly anoxic soils, or at the oxic-anoxic transition, is needed.

### 7.1.2 Fe may affect priming by shaping microbial community composition

Fe oxides may alter microbial community composition and soil C and N content (Heckman et al., 2009; Heckman et al., 2018), likely through controlling nutrient availability and affecting the structural properties of dissolved organic matter (DOM). For instance, the application of goethite to soil limits P and N bioavailability and increases the aromatic content of water extractable organic matter (WEOM), which may lower the ratio of fungi to bacteria (Heckman et al., 2012). In general, fungi have low C use efficiency (CUE) (Silva-Sánchez et al., 2019) and are associated with efficient N cycling (Wardle et al., 2004). Unlike bacteria, fungi require less N per unit biomass which may result in decreased N mining from MAOM. In addition, applying goethite together with arbuscular mycorrhizal fungi (AMF) to soil decreased priming in the rhizosphere by protecting OM through sorption and aggregate formation by AMF hyphae (Jeewani et al., 2021b). Godbold et al., (2006) hypothesized that the turnover of the mycorrhizal external mycelium is the dominant process by which root-derived C is incorporated into stable SOM pools (Godbold et al., 2006) and distributed throughout the soil (Frey, 2019), which contribute to SOM sequestration (Godbold et al., 2006). However, mycorrhizal fungi can destabilize SOM by multiple mechanisms summarized in Frey (2019). For example, N in MAOM can be made available to plants by mycorrhizal hyphae which extend plant roots deeper in soils and may destabilize aggregates that protect MAON (Jilling et al., 2018).

### 7.2 Birch effect

The Birch effect is defined as a short-term pulse in C and N mineralization caused by soil drying and rewetting. Although many studies have been done on N mineralization and nitrification (Birch, 1958, 1959, 1960, 1964; Wilson and Baldwin, 2008), the studies on the Birch effect have mainly focused on C. A pattern has been observed was that N mineralization rate increases as soil becomes drier, along with a rapid decline when soil is rewetted. Soil moisture is accompanied by increased $NO_3^-$ production. The origin of this pattern remains elusive, though the Birch effect is generally tied to multiple interacting mechanisms, including the dissolution of organo-mineral bonds, which increases the accessibility of substrates to microbial degradation.

Wilhelm et al., (2022) investigated the effects of wet-dry cycles on C mineralization of newly added substrates in soils with different Fe and SOC contents and developed under different precipitation regimes. The authors found that wet-dry cycles did not affect C mineralization in the ferrihydrite-rich soil, due to C substrates being incorporated into microbial biomass and their stabilization in newly formed Fe-organic associations. In contrast, soils with low Fe content did not have enough available surfaces to form Fe-organic associations. Thus, C substrates were more susceptible to mineralization mediated by wet-dry cycles in these soils (Wilhelm et al., 2022). The availability of reactive Fe surfaces in soils can therefore decrease the mineralization of newly formed C during wet-dry cycles.





In tropical regions, soils are widely dominated by Fe oxides that sorb SOC but are also subjected to rapid redox-induced mineral transformations due to highly dynamic wet-dry cycles. In fact, the transformation of amorphous Fe oxides into more crystalline forms decreases soil sorption capacity and nutrient retention (Attygalla et al., 2016; Wilmoth et al., 2018; Chen et al., 2020). We hypothesize that wet dry-cycles can induce rapid electron transfer from and to Fe oxides, known as cryptic Fe cycle, which may affect N bioavailability. During the wet period, Fe(III) oxides can be used as an electron acceptor and be

reduced to Fe(II), which can abiotically react with $NO_3^-$ to form $NH_4^+$, or with nitrite ($NO_2^-$) to form $N_2O$. This Fe(II) can be converted back to Fe(III) oxides during the dry period, which may sorb OM and protect it against further degradation or generate oxidative radicals through Fenton reactions that break down organics, including N compounds. This cryptic cycling of Fe will have a varied effect on the role of Fe in controlling N bioavailability over short spatiotemporal scales, which may either increase or decrease bioavailable N. Further research is needed to detangle these interactions.

**7.3 Fe in the context of freeze-thaw cycles: the case of permafrost-affected soils**

Permafrost-affected soils store large amounts of OC and ON as a result of SOM stabilization due to freezing of SOM and cryoturbation. Along a permafrost soil chronosequence, Joss et al., (2022) found a high percentage of FeOM in cryoturbated soils compared to organic or mineral horizons. Cryoturbation also favors the accumulation of SOM with high C:N ratio at deeper soil depths (Treat et al., 2016a), which also may be present as associations with Fe minerals or in particulate organic

matter. Upon thawing, this tremendous amount of SOC and total nitrogen (TN) facilitate high gross N turnover rates by heterotrophic processes. For instance, Treat et al.(2016b) observed increased nitrogen availability during long thaw seasons in tundra soils, whereas other authors reported higher $N_2O$ emissions from increased denitrification (Cui et al., 2016; Yang et al., 2016; Yang et al., 2018). This is partly because SOC and SOM, previously trapped in FeOM associations, are released and exposed to microbial degradation (Harden et al., 2012; Gentsch et al., 2015; Mueller et al., 2015; Patzner et al., 2020). In fact,

Patzner et al., (2020) found that along a thaw gradient, the amount of dissolved organic carbon (DOC) increased as well as the abundance of Fe(III)-reducing bacteria which use Fe(III) as terminal electron acceptor and oxidize OM. The importance of this mechanism in N destabilization likely depends on the extent to which Fe dissolution contributes to soil OM persistence in redox-dynamic permafrost (Patzner et al., 2020). More investigations of Fe control on N bioavailability in permafrost-affected soils are needed, especially with the recent development pointing out that mineral N cycling is as important as ON cycling in

the active layers of these soils (Ramm et al., 2022).

**8 Impact of global change on Fe-N bioavailability interactions**

Global change affects Fe-N interactions in multiple ways. First, climate change is expected to increase the occurrence of the Birch effect as a result of extreme variability in precipitation, which affects N bioavailability. Fe plays multiple roles in this process; Fe can protect ON from decomposition in drier soils but its reaction with light can lead to Fenton-reaction induced

ON decomposition (Georgiou et al., 2015). In wetter soil, ON destabilization rates can increase as a result of fluctuations in





redox conditions, the occurrence of cryptic Fe cycling and modifications of mineral properties. Second, climate change lead to elevated atmospheric $CO_2$ concentration ($eCO_2$), but the effects of the latter on Fe-N bioavailability interactions are not well understood. Recent research showed that $eCO_2$ stimulates root and microbial respiration, which can decrease soil redox potential causing Fe reduction to proceed (Cheng et al., 2010). The production of Fe(II), which increased by 64% under $eCO_2$

treatment, caused substantial losses of $NH_4^+$ via Feammox in a 15-year free-air $CO_2$ enrichment (FACE) study in rice paddy systems. Feammox was meditated by autotrophic anaerobes that may use soil $CO_2$ as C source to couple anaerobic ammonium oxidation and Fe reduction (Xu, C. et al., 2020). $eCO_2$ can also increase the destabilization of MAON via priming, as, $eCO_2$ increases root biomass and associated exudate production at deeper soil depths, enabling the liberation of large amount of deep soil N from these associations (Iversen, 2010). This increased turnover of N from MAOM would probably be substantial under

future $eCO_2$. Third, land use change involving the conversion to agriculture can decrease SON (García-Oliva et al., 2006). We hypothesize that this decline in SON is influenced by the effects of land use change on Fe cycling. For example, it was observed that the crystallinity of Fe oxides increased when forests were converted to agricultural fields in the Southern Piedmont, USA (Li and Richter, 2012). Additionally, Tan et al., (2019) showed that land use change from fallow to paddy soils promoted Fe reduction by decreasing soil pH and increasing the electron shuttling capacity of SOM due to increased organo-Fe associations

(Tan et al., 2019), which may accelerate N turnover by processes such as Feammox. Fourth, freeze-thaw cycles are expected to increase due to climate change. Warmer temperatures increase permafrost thaw which may increase redox-meditated heterotrophic N turnover processes and the destabilization of FeON. To conclude, global change affects the roles of Fe in N bioavailability which may in turn affect the balance between Fe-meditated SON destabilization and protection.

## 9 Synthesis and outlook

Attempts at understanding controls and drivers of N bioavailability, a fundamental soil ecosystem property, often omit the role of Fe minerals. However, the tendency of proteins to associate strongly with minerals, and the involvement of the latter in both enzymatic and non-enzymatic reactions that influence the N cycle has motivated this review, which specifically focuses on Fe-N bioavailability interactions (Fig. 4). Including Fe in current models of SOM is challenging because the mechanisms by which Fe controls N storage, stabilization, bioavailability, and loss are complex and remain incompletely understood. This is

because the present knowledge is, on one hand, based on OM-mineral correlations, which is a simplistic approach since correlations tend to be specific for certain soil conditions and types (Kleber et al., 2021; Wagai et al., 2020), and on the other hand, knowledge is impeded by limitations in the analytical framework used to explore these interactions. In this section, we highlight challenges and opportunities for future research.



**Figure 4. Fe affects N bioavailability in soils. This Fig. doesn't specify soil conditions under which an Fe role may proceed.**

**9.1 Sorbent role of Fe in controlling N bioavailability**

The sorbent role of Fe in controlling N bioavailability is multifaceted. Sorption can protect N from decomposition by reducing the activity of enzymes and limiting the accessibility of N substrates to degradation mechanisms. However, a fraction of sorbed N is bioavailable (Bird et al., 2002; Kleber et al., 2007), or can be made available by processes such as priming or displacement

by competitive organics. Thus, the concept of "sorptive stabilization" of N substrates does not stand as a conclusive explanation for N persistence in soils and should rather be revisited. In this context, sorption to Fe minerals may impose spatial constraints on the accessibility of N substances to microbes, as sorption can locate N in physically isolated spaces such as micropores,





microaggregates, or microdomains of densely arranged clays which slows down its decomposition and decreases its bioavailability (Kleber et al., 2021).

Research on Fe-meditated N depolymerization has mostly focused on proteins (Wanek et al., 2010; Noll et al., 2019; Reuter et al., 2020), since proteins alone constitute 60% or more of the N in plant and microbial cells (Fuchs, 1999) and are strongly sorbed to Fe surfaces. However, not all soil and mineral-associated N is protein. Rather, N exists in a variety of chemical forms (box 1) including microbial cell wall compounds. Using Fourier transform infrared spectroscopy (FTIR) and isotope pool dilution (IPD), multiple studies have shown the importance of microbial cell wall depolymerization in the delivery of soil N

(Hu et al., 2017; Hu et al., 2018; Hu et al., 2020). In addition, depolymerization of membrane lipids and nucleic acids is not yet characterized despite the detection of their degradation products in soils (Warren, 2021). This leads to the following question: how important is the chemical form of Fe-associated N in determining soil N bioavailability? This is relevant since the molecular characteristic of different N forms influences the type and strength of bonding with minerals, which may affect N bioavailability. For instance, Fe oxyhydroxides binds amino sugars more strongly than proteins in boreal forests (Keiluweit

and Kuyper, 2020), likely allowing less mineralization from the former compared to the latter  compounds.

### 9.2 Structural role of Fe in controlling N bioavailability

Despite a small number of studies relating structural Fe in clays and aggregates to N bioavailability, the dynamics of these interactions and relevant mechanisms remain elusive. Several questions remain to be resolved, including: are the original structure and physico-chemical characteristics of clay minerals restored upon reoxidation of its structural Fe? If so, what are

the implications for $NH_4^+$ release and fixation and other processes that influence loss and gain of bioavailable N? How relevant is the loss of Fe by solubilization and reduction to microaggregate instability and N bioavailability in soils? In addition, the relevance and the occurrence of Fe-induced C and N polymerization is soils awaits confirmation, because this phenomenon has been observed only in laboratory settings.

### 9.3 The role of Fe as a catalyst in controlling N bioavailability

Assessing the importance of Fe-mediated ROS generation in N bioavailability is a formidable challenge.  In fact, despite being common in soils, ROS have extremely short lifetimes and are highly reactive towards other soil constituents such as carbonates and bromide (Kleber et al., 2021), which complicate their detection in soils. They are produced by both abiotic and biotic pathways, and the contribution of each pathway to N bioavailability remains elusive. Additionally, rates and mechanisms of ROS production from these two pathways are still not known. Such information is particularly important to understand N

dynamics in environments conducive to ROS formation, such as oxic/anoxic zones, environments with intense solar radiation or in boreal forests where fungi use ROS based mechanisms to access Fe-sorbed N. In contrast to their decomposition role, Yu et al.(2020) found an important role of Fe-meditated ROS production in OM polymerization, which increases the recalcitrance of OM and its resistance to degradation mechanisms (Yu et al., 2020). This finding sheds light on other controls and pathways relevant to N bioavailability. For example, under what conditions can the role of Fe-mediated ROS generation on N





bioavailability be shifted from decomposition to protection? And how will this evolve in a changing world where solar radiation is becoming more intense and the frequencies of extreme events (e.g., droughts, rain) is increasing?

**9.4 Electron transfer role of Fe in controlling N bioavailability**

The capacity of Fe to act as an electron acceptor and donor can affect bioavailable N loss from soils by processes such as Feammox and denitrification. To further understand these processes, more research is needed on cryptic Fe cycling and the
controls over the oxidation-reduction dynamics of Fe in soil, since preservation of oxidized Fe promotes N stabilization within mineral associations. For instance, the effects of added electron shuttles on the extent and the rate of Fe(III) reduction and associated loss of N via Feammox have been investigated, however, the capacity of SOM and organo-Fe associations to transfer electrons has received less attention (Sposito, 2011; Xu, Z. et al., 2020). The characterization and mapping of spatiotemporal redox heterogeneity also deserves attention (Wilmoth, 2021).

**9.5 Varied analytical approach is needed to characterize Fe-N interactions**

To understand the roles of Fe in controlling N bioavailability, a varied analytical approach must be adopted to enable a more holistic and multidimensional view of these interactions, considering all the possible outcomes of Fe reactions on N as driven by the physico-chemical and biological characteristics of soil and management. This approach is essential to provide realistic turnover rates of N and decipher the underlying mechanisms of Fe-N reactions in soil, in contrast to controlled lab experiments
which do not represent soil in its complexity and heterogeneity. This approach should also capture variations in the processes of interest within multiscale and time dimensions. Here, we present most common and powerful techniques that can be combined in the framework of this varied approach to understand Fe-N interactions. Note that an extensive list of techniques is out of the scope of this review.

(a)   Imaging techniques: Techniques such as Synchrotron XAS and Synchrotron X-ray allow the identification and the
535        characterization of structural and chemical properties of minerals as well as their oxidation states. They can also be used to determine the speciation of SON and dissolved organic nitrogen (DON) as well as the structural characteristics of soil, such as pore size and pore connectivity. These information help, for example, to characterize the fine-scale redox heterogeneity (Wilmoth, 2021) that affects Fe cycling and its interconnection with N bioavailability. In addition, these techniques are used to observe and investigate the 3D structure of organo-Fe minerals in soils. Kleber
540        et al. (2021) called for using them in studies of enzyme activity because they allow the investigation of the natural structure of organo-mineral associations without alteration (Kleber et al., 2021). However, while using advanced imaging techniques reveals information at fine scales, upscaling such data is challenging (Wagai et al., 2020).

(b)  Microbial techniques: They provide information on the identity of microbial taxa regulating soil biogeochemical processes in question. They include techniques such as metatranscriptomics which can be used to distinguish the
545        biological from the abiotic pathways used to direct redox reactions (Wilmoth, 2021), and metagenomics that were used recently to explore coupled nutrients interactions, including coupled Fe-N reactions (Ma et al., 2021).

(c) Isotope techniques: Isotopes can be used to determine gross rates and the investigation of the pathways and mechanisms of the processes in question. They can also be used to determine OM pools with varying turnover rates. Stable isotope probing, which is a high-resolution technique, can also be used to trace the microbial uptake of N as affected by Fe minerals as well as its fate in soil environments.

(d) Molecular characterization techniques: These techniques, which include FTIR, allow the identification of different soil organic molecules and the analysis of their bonding mode and strength with minerals.

## 9.6 Concluding Comment

As a final commentary on Fe-N bioavailability interactions, we propose the following questions: how much N can be mobilized by Fe-related mechanisms? What are the controls on these interactions? And how important are certain mechanisms relative to others in securing N bioavailability in the context of global change? Do reactions observed in laboratory settings occur naturally in soils? We also urge the field to develop new methods and techniques, such as those capable of detecting low concentrations of ROS and their fate in soil environment, or the products of mineral-induced OM polymerization.

## Author contributions

IS conducted the literature review and wrote the manuscript, XZB, PL and WH proofread, edited, reviewed, provided guidance and advice on manuscript development.

## Competing interests

The authors declare that they have no conflict of interest.

## Acknowledgements

We acknowledge funding from Mohammed VI Polytechnic University and the J. G. Boswell Endowed Chair in Soil Science.

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
