# Peer review of "Reviews and syntheses: Iron: A driver of nitrogen bioavailability in soils?"

_Biogeosciences, 2022_

## Author Comment (AC1)

We greatly appreciate the amount of time and effort put into this review as evidenced by the extremely constructive comments provided! We will address the reviewer's concerns by reorganizing, structuring, re-writing, and summarizing the text in the manuscript as described below.

1. Lines 33-52: This list of evidence reflects the structure of the paper overall – many sections in the paper are stand alone "chunks" of ideas that do not cohesively tie together. As written, the sections appear as a list of ideas instead of a defined structure with a beginning, middle, and end.

Response: We agree with this helpful comment. This list is transformed into a paragraph with a beginning, middle and an end. It reads now as follows:

Line 30: "Since C and N cycles are interconnected in soils (Feng et al., 2019; Gärdenäs et al., 2011), they should be regulated by the same factors, including mineralogy type (Wade et al., 2018). Increasing evidence shows that Fe specifically represents a major control over N biological transformations, including mineralization (Wade et al., 2018), nitrification (Huang et al., 2016a) (Han et al., 2018) denitrification (Zhu et al., 2013) (Wang et al., 2016), as well as their abiotic analogous reactions, such as chemo-denitrification (Burger and Venterea, 2011) and Fe-mediated hydroxylamine oxidation to nitrous oxide ($N_2O$). These reactions and others (Fig.1) are likely to operate ubiquitously in soils, due to the close proximity between Fe minerals and SOM since most of the latter is contained in association with the former (Lalonde et al., 2012; Wagai and Mayer, 2007)".

2. Section 4: It is not clear how the structural role is distinct from the sorbent role. The mechanisms presented in Figure 2 and many described in this section are referring to adsorption/desorption processes.

Response: The purpose of this section is to highlight the role of Fe in the formation and stability of micro-aggregates and the impact this has on N bioavailability. This section refers to the fact that Fe-mediated aggregate stability increases N stability inside microaggregates by limiting its physical accessibility to microbes. The intent was to highlight a physical rather than a chemical phenomenon, therefore, we eliminated all the text referring to sorption-adsorption mechanisms. Thank you so much for bringing this to our attention, that was a great review comment!

**This section now reads as:

Fe oxides are one of the most important constituents of soil microaggregates (Peng et al., 2015), serving as nuclei for their formation and meditating their stability (Barral et al.,

1998; Pronk et al., 2012; Wei et al., 2016), acting as a cementing agent (Colombo and Torrent, 1991; Krause et al., 2020) and binding OM (Giovannini and Sequi, 1976; Totsche et al., 2017). Recent studies demonstrated that colloidal-sized Fe promotes the formation of smaller-sized microaggregates (<20 μm) and Fe-rich biosolids induce rapid formation of microaggregates and significantly increase soil organic carbon (SOC). (Krause et al., 2020) (Silva et al., 2015).N is also an important component of microaggregate-SOM (Golchin et al., 1994) (Aufdenkampe et al., 2001) (Sollins et al., 2006). Using density fractionation, Wagai et al., (2020) observed joint accumulation of OM with low C:N ratio and pedogenic Fe and Al oxides in the meso-density fractions (1.8–2.4 $\text{g cm}^{-3}$) of five soil orders collected from different climate zones. Moreover, Rodionov et al., (2001) observed high concentrations of amino sugars in microaggregates (Rodionov et al., 2001). These observations have implications for N bioavailability, given the fact that Fe mediated micro-aggregation may slow down or suppress N mineralization (Mendes et al., 1999). Indeed, N compounds located inside microaggregates have lower availability to microbes than those located on more accessible surfaces. Microaggregate-N is also relatively more persistent than macroaggregate-N because microaggregates' turnover is relatively slow, which provides longer-term stabilization of OM (Cambardella and Elliott, 1993; Six et al., 2002). Similarly, Bugeja and Castellano (2018) observed positive correlation between ammonium oxalate-extractable Fe (AmOx-F), C and N in microaggregate, indicating that Fe and microaggregate stabilization are interconnected (Bugeja and Castellano, 2018). Partial or complete removal of mineral-forming components, for example due to Fe reduction, can initiate aggregate turnover and destabilization (Michalet, 1993; Cornell and Schwertmann, 2003) which eventually exposes OM to microbial degradation (Lützow et al., 2006) and organic carbon (OC) and ON loss from SOM (Cambardella and Elliott, 1993). We also hypothesize that there is another pathway by which Fe-promoted aggregation may decrease N mineralization. Aggregates of different sizes influence microbial community composition differently and therefore the activities of N mineralization enzymes (Muruganandam et al., 2009). Therefore, it will be useful to examine the distribution and the activities of these enzymes among soil aggregate size classes along a gradient of increased Fe mineral content in soils.

We also updated Figure 2:

[Figure]

Figure: Schematic representation of the effects of Fe-promoted aggregate formation and stability on N accessibility to microbial degradation.

3. Sections 7 and 8, in particular, lack an overall structure. A potential solution to the organizational issues with the writing would be to separate the properties and processes into distinct spatial scales: 1) The molecular scale at which sorption/desorption, catalysis, electron transfer occurs, 2) the micro-scale, at which iron mediates soil aggregation, and 3) the meso/ecosystem-scale, at which iron may influence the priming of soil nitrogen in the rhizosphere or the response of SON cycling to global change.

**Response:** We thank the reviewer for this helpful and well thought of comment. We love the idea of separating processes/mechanisms into scales, but we didn't feel that it would follow the flow of the narrative and may make the review too long. To make the structure clearer to the reader, we decided to introduce section 7 and 8; section 7 which details the role of Fe in the three complex phenomena that affects N bioavailability in soils; priming, birch effect and freeze-thaw cycle and section 8 with the focus on how is Fe-N bioavailability influenced by global change.

\*\*The paragraph in the introduction now reads as:

line 70: 'While these roles of Fe in controlling C cycling have been studied extensively, their effects on N bioavailability are not well explored. This review seeks to underpin these suggested relationships and provide mechanistic descriptions of how Fe controls N bioavailability in soils. Moreover, we detail how Fe participates in three complex

phenomena that influence N bioavailability; priming, birch effect, and freeze-thaw cycle. We also highlight how Fe-N interactions are affected by global change. This information are needed to construct reliable models with improved predictive power of N cycling in terrestrial ecosystems (Wade et al., 2018), and will offer new possibilities for land management".

**We also included an introductory paragraph at the beginning of section 7 where we talk about how the three phenomena influence N bioavailability and why the role of Fe should be examined.

**To better structure section 8, we synthesized its information and introduced subheads to highlight the driver of change and potential impact on Fe-N bioavailability. We also included an introductory paragraph to this section.

This section subheads are:

**8 Impact of global change on Fe-N bioavailability interactions**

8.1 Impact of variability in precipitation

8.2 Impact of variability in temperature

8.2 Impact of elevated $CO_2$

8.3 Impact of land use change

We hope these changes are satisfactory and address the reviewer concerns.

---

## Author Comment (AC2)

We greatly appreciate the amount of time and effort put into this review as evidenced by the extremely constructive comments provided! We will address the reviewer's concerns by reorganizing, structuring, re-writing, and summarizing the text in the manuscript as described below.

**We profoundly thank the reviewer for providing a list of relevant and excellent reviews on N bioavailability. These reviews have been read and incorporated into the introduction of this manuscript to support the need to study Fe minerals impact on N bioavailability.

1. L33-51. This section might read better as prose rather than a list that doesn't seem to be ordered in a particular way.

Response: We agree with this helpful comment. This list is transformed into a paragraph with a beginning, middle and an end. It reads now as follows:

Line 30: "Since C and N cycles are interconnected in soils (Feng et al., 2019; Gärdenäs et al., 2011), they should be regulated by the same factors, including mineralogy type (Wade et al., 2018). Increasing evidence shows that Fe specifically represents a major control over N biological transformations, including mineralization (Wade et al., 2018), nitrification (Huang et al., 2016a) (Han et al., 2018) denitrification (Zhu et al., 2013) (Wang et al., 2016), as well as their abiotic analogous reactions, such as chemo-denitrification (Burger and Venterea, 2011) and Fe-mediated hydroxylamine oxidation to nitrous oxide ($N_2O$). These reactions and others (Fig.1) are likely to operate ubiquitously in soils, due to the close proximity between Fe minerals and SOM since most of the latter is contained in association with the former (Lalonde et al., 2012; Wagai and Mayer, 2007)".

2. L53-70. This section provides a nice structure for the paper.

Response: Thank you for this helpful comment. We like this structure too !

3. Fig. 1. Is this an original figure or taken from another source? If the latter, the source needs to be provided. If the former, I would suggest using color to make it more interesting/readable and also link it to the overarching structure of the paper. That is, if you're going to follow the structure outlined in L53-70, then indicate how these four "roles" tie into the figure. Or develop a different overarching figure that gives a conceptual overview of the paper.

Response: Thank you very much for bringing this to our attention. This figure was adapted from (Zhu-Barker et al. 2016). The original figure from Zhu-Barker et al. did not have all the processes that I added to it. We felt adding the 4 roles of Fe to this figure would make it cluttered and the purpose was just to show that Fe takes part in almost every N process. So, we decided to develop another figure to give a conceptual overview of the paper (as per the recommendation). See attached.

[Figure]

4. Sec. 2. This section interrupts the flow from the overarching structure of the paper laid out in L53-70 and Sec. 3 (i.e., "sorbent role"). Perhaps this material could be shortened and included in the intro section before laying out the paper objectives/structure

We agree with this helpful comment. Material has been shortened and added to the introduction of this manuscript. This section reads now as:

**Starting at the line 30:** Since C and N cycles are interconnected in soils (Feng et al., 2019; Gärdenäs et al., 2011), they should be regulated by the same factors, including mineralogy type (Wade et al., 2018). Increasing evidence shows that Fe specifically represents a major control over N biological transformations, including mineralization (Wade et al., 2018), nitrification (Huang et al., 2016a) (Han et al., 2018) denitrification (Zhu et al., 2013) (Wang et al., 2016), as well as their abiotic analogous reactions, such as chemo-denitrification (Burger and Venterea, 2011) and Fe-mediated hydroxylamine oxidation to nitrous oxide ($N_2O$). These reactions and others Fig.1, are likely to operate ubiquitously in soils, due to the close proximity between Fe minerals and SOM since most of the latter is contained in association with the former (Lalonde et al., 2012; Wagai and Mayer, 2007).

The characteristic properties of individual Fe minerals and N compounds and how these properties are influenced by the soil environment likely drive the aforementioned reactions as well. First, Fe exists in a variety of polymorphs (Navrotsky et al., 2008) and is a redox-sensitive element that cycles between Fe(II) and Fe(III) states as controlled by soil Eh and pH. While Fe(III) promotes N stabilization within mineral associations, Fe(III) mobilization when it is reduced to Fe(II) can release N into solution. Fe reactivity is also driven by the amount and sign of surface charge, surface topography, particle size, crystallinity (Li et al., 2015a) (Petridis et al., 2014) and the presence and the type of organic matter (OM) coverage (Gao et al., 2018; Kleber et al., 2007; Poggenburg et al., 2018) (Boland et al., 2014; Henneberry et al., 2016) (Kaiser and Zech, 2000a) (Daugherty et al., 2017). Second, soil N exists predominantly in organic forms (ON); mostly as protein and peptides, and to a lesser extent as amino-sugars and nucleic acids (Kögel-Knabner, 2006) (Knicker, 2011; Schulten and Schnitzer, 1997). Proteins are intrinsically reactive towards soil minerals, due to a number of properties, including hydrophobicity, surface charge distribution, surface area, number and type of functional groups, conformation, and size (Lützow et al., 2006). N from these compounds is generally not directly bioavailable due to molecular size constraints on microbial cell uptake (Schimel and Bennett, 2004). Depolymerization reactions, carried out by the activity of extracellular enzymes, such as peptidases, transform these polymers into soluble, low molecular weight (MW) organic monomers (e.g., short oligopeptides, amino acids (AAs)). Recent research shows that the size of AAs available for mineralization is controlled by peptidase activity, but more so by protein availability, both of which are affected by the interactions with Fe minerals. Therefore, Fe may drive gross AA production in soils (Noll et al., 2019).

5.  L127. This question doesn't lead well into the overarching theme of this section and detracts from the main thread. Start with an introductory paragraph that

introduces the topic and provides a roadmap for the rest of the section. I assume that the "sorbent role" isn't just about enzymes, right?

**Response:** We agree with this helpful comment. An introductory paragraph has been added to all the sections.

For section 1 for example, this paragraph reads as: "This section explores the role of Fe in regulating N bioavailability through sorption processes, which alter both the availability of N substrates and enzyme activity. This section also highlights the multiple mechanisms of destabilization of sorbed N; such as Fe reduction, desorption by local disequilibrium in soil chemistry and the displacement of N by competitive sorption".

6. Sec. 4. As noted above, starting the section with a question is not the best way to introduce the topic and provide an overview for the section. In the first paragraph, lay out what the questions or topics are and then follow up with a succinct discussion of each. Ditto this comment for all sections.

**Response:** We agree with this helpful comment. An introductory paragraph has been added to all the sections. For section 4, this paragraph reads as:

"The impact of structural Fe on N bioavailability in soils is a complex phenomenon that can be influenced by various factors. This section provides a detailed examination of this subject, focusing on how the structural Fe in clay and aggregates influence N turnover, as well as Fe- induced organic nitrogen (ON) polymerization"

7. Fig. 2. I would encourage you to think about how this figure could be a bit more nuanced rather than just having "clouds" for enzymes, N substrates, etc.

Response: This figure has been eliminated from the manuscript to address the first reviewer comment. An updated figure has been incorporated into the manuscript (see attached).

[Figure]

Figure: Schematic representation of the effects of Fe-promoted aggregate formation and stability on N accessibility to microbial degradation.

8.  Sec. 5. You skip the "electron transfer role" and go straight to "catalytic role". If there isn't a separate section for "electron transfer role" then remove from overview paragraph as a separate, defined "role".

Response: The electron Transfer role of Fe is presented in lines 325-350

9.  Sec. 7 and 8. As noted by the other reviewer, these sections lack structure and were not adequately introduced earlier in the manuscript.

Response: We thank the reviewer for this helpful and well thought of comment. We love the idea of separating processes/mechanisms into scales, but we didn't feel that it would follow the flow of the narrative and may make the review too long. To make the structure clearer to the reader, we decided to introduce section 7 and 8; section 7 which details the role of Fe in the three complex phenomena that affects N bioavailability in soils; priming, birch effect and freeze-thaw cycle and section 8 with the focus on how is Fe-N bioavailability influenced by global change.

**The paragraph in the introduction now reads as:

line 70: 'While these roles of Fe in controlling C cycling have been studied extensively, their effects on N bioavailability are not well explored. This review seeks to underpin these suggested relationships and provide mechanistic descriptions of how Fe controls N bioavailability in soils. Moreover, we detail how Fe participates in three complex phenomena that influence N bioavailability; priming, birch effect, and freeze-thaw cycle. We also highlight how Fe-N interactions are affected by global change. This information are needed to construct reliable models with improved predictive power of N cycling in terrestrial ecosystems (Wade et al., 2018), and will offer new possibilities for land management".

**We also included an introductory paragraph at the beginning of section 7 where we talk about how the three phenomena influence N bioavailability and why the role of Fe should be examined.

**To better structure section 8, we synthesized its information and introduced subheads to highlight the driver of change and potential impact on Fe-N bioavailability. We also included an introductory paragraph to this section.

This section subheads are:

**8 Impact of global change on Fe-N bioavailability interactions**

8.1 Impact of variability in precipitation

8.2 Impact of variability in temperature

8.2 Impact of elevated $CO_2$

8.3 Impact of land use change

All the figures of the manuscript have been updated with colors and revised. Here are examples:

[Figure]

[Figure]

We hope these changes are satisfactory and address the reviewer's concerns.

---

## Author Response (AR2)

We thank the reviewer for their comments on this version of the manuscript.

The manuscript was checked for all the helpful suggestions and corrections that were minor.

We just shed light that we changed the information in line 400 to "Beyond SOM stabilization, Fe oxides may regulate priming by altering microbial community composition, soil C and N content (Heckman et al., 2009; Heckman et al., 2018); potentially by restricting nutrient availability and changing the structural properties of dissolved organic matter (DOM). For instance, the application of goethite to soil limits P and N bioavailability while increasing the aromatic content of water extractable organic matter (WEOM), which may lower the ratio of fungi to 400 bacteria (Heckman et al., 2012) and alter C and N cycling as a consequence (Silva-Sánchez et al., 2019; Wardle et al., 2004)." To account for the fact that fungi do not necessarily have low CUE compared to bacteria. Thank you very much for this comment.

We hope this version of the manuscript is successful.

---

## Author Response (AR3)

**Response to reviewer 1:**

We greatly appreciate the amount of time and effort put into this review as evidenced by the extremely constructive comments provided! We will address the reviewer's concerns by reorganizing, structuring, re-writing, and summarizing the text in the manuscript as described below.

1. **Lines 33-52: This list of evidence reflects the structure of the paper overall – many sections in the paper are stand alone "chunks" of ideas that do not cohesively tie together. As written, the sections appear as a list of ideas instead of a defined structure with a beginning, middle, and end.**

**Response:** We agree with this helpful comment. This list is transformed into a paragraph with a beginning, middle and an end. It reads now as follows: Line 40: "Since C and N cycles are interconnected in soils (Feng et al., 2019; Gärdenäs et al., 2011), they should be regulated by the same factors, including mineralogy type (Wade et al., 2018). Increasing evidence shows that Fe specifically represents a major control over N biological transformations, including mineralization (Wade et al., 2018), nitrification (Huang et al., 2016a) (Han et al., 2018) denitrification (Zhu et al., 2013) (Wang et al., 2016), as well as their abiotic analogous reactions, such as chemo-denitrification (Burger and Venterea, 2011) and Fe-mediated hydroxylamine oxidation to nitrous oxide (N2O). These reactions and others (Fig.1) are likely to operate ubiquitously in soils, due to the close proximity between Fe minerals and SOM since most of the latter is contained in association with the former (Lalonde et al., 2012; Wagai and Mayer, 2007)".

2. **Section 4: It is not clear how the structural role is distinct from the sorbent role. The mechanisms presented in Figure 2 and many described in this section are referring to adsorption/desorption processes.**

**Response:** The purpose of this section is to highlight the role of Fe in the formation and stability of micro-aggregates and the impact this has on N bioavailability. This section refers to the fact that Fe-mediated aggregate stability increases N stability inside microaggregates by limiting its physical accessibility to microbes. The intent was to highlight a physical rather than a chemical phenomenon, therefore, we eliminated all the text referring to sorption-adsorption mechanisms. Thank you so much for bringing this to our attention, that was a great review comment! We also updated the figure associated with this section (line 190).

We also removed the section on Fe present in clay and N bioavailability because it was purely a sorption mechanism. The sorbent role of Fe in controlling N bioavailability contains two portions now, one on organic N (line 130-240) and one on inorganic N (line 240-250). The latter portion is concerned with the role of Fe in controlling inorganic N bioavailability with the example of how Fe present in clay in affecting is affecting it.

3. **Sections 7 and 8, in particular, lack an overall structure. A potential solution to the organizational issues with the writing would be to separate the properties and processes into distinct spatial scales: 1) The molecular scale at which sorption/desorption, catalysis, electron transfer occurs, 2) the micro-scale, at which iron mediates soil aggregation, and 3) the meso/ecosystem-scale, at which iron may influence the priming of soil nitrogen in the rhizosphere or the response of SON cycling to global change.**

**Response:** We thank the reviewer for this helpful and well thought of comment. We love the idea of separating processes/mechanisms into scales, but we didn't feel that it would follow the flow of the narrative and may make the review too long. To make the structure clearer to the reader, we decided to introduce section 7 and 8; section 7 which details the role of Fe in the three complex phenomena that affects N bioavailability in soils; priming, birch effect and freeze-thaw cycle and section 8 with the focus on how is Fe-N bioavailability influenced by global change.

Since we rewrote the whole document, the numbering and subheadings have been reimagined to accommodate new changes. This is the overall structure of the document:

1. Introduction
2. Roles of Fe in controlling N bioavailability
3. Involvement of Fe in soil phenomena that affect N bioavailability
4. Impact of global change on Fe-N bioavailability interactions
5. Synthesis and outlook

Each section has an introductory paragraph to guide the reader through the information provided within that section.

Here is an example at the line 450:

**4 Impact of global change on Fe-N bioavailability interactions**

Anticipated future climate scenarios indicate substantial fluctuations in precipitation and temperature patterns, accompanied 450 by increasing levels of atmospheric CO2. These changes, along with alterations in land use, have the potential to significantly impact Fe-N bioavailability interactions in various ways, which are detailed below.

**Responses to reviewer 2:**

We greatly appreciate the amount of time and effort put into this review as evidenced by the extremely constructive comments provided! We will address the reviewer's concerns by reorganizing, structuring, re-writing, and summarizing the text in the manuscript as described below.

1. **L33-51. This section might read better as prose rather than a list that doesn't seem to be ordered in a particular way. Response: We agree with this helpful comment.**

This list is transformed into a paragraph with a beginning, middle and an end. It reads now as follows: Line 40: : Line 40: "Since C and N cycles are interconnected in soils (Feng et al., 2019; Gärdenäs et al., 2011), they should be regulated by the same factors, including mineralogy type (Wade et al., 2018). Increasing evidence shows that Fe specifically represents a major control over N biological transformations, including mineralization (Wade et al., 2018), nitrification (Huang et al., 2016a) (Han et al., 2018) denitrification (Zhu et al., 2013) (Wang et al., 2016), as well as their abiotic analogous reactions, such as chemo-denitrification (Burger and Venterea, 2011) and Fe-mediated hydroxylamine oxidation to nitrous oxide (N2O). These reactions and others (Fig.1) are likely to operate ubiquitously in soils, due to the close proximity between Fe minerals and SOM since most of the latter is contained in association with the former (Lalonde et al., 2012; Wagai and Mayer, 2007)".

2. **L53-70. This section provides a nice structure for the paper.**

**Response:** Thank you for this helpful comment. We like this structure too !

3. **Fig. 1. Is this an original figure or taken from another source? If the latter, the source needs to be provided. If the former, I would suggest using color to make it more interesting/readable and also link it to the overarching structure of the paper. That is, if you're going to follow the structure outlined in L53-70, then indicate how these four "roles" tie into the figure. Or develop a different overarching figure that gives a conceptual overview of the paper.**

 **Response:** Thank you very much for bringing this to our attention. This figure was adapted from (Zhu-Barker et al. 2016). The original figure from Zhu-Barker et al. did not have all the processes that I added to it. We felt adding the 4 roles of Fe to this figure would make it cluttered and the purpose was just to show that Fe takes part in almost every N process. So, we decided to develop another figure to give a conceptual overview of the paper (as per the recommendation). It is at line 105.

4. **Sec. 2. This section interrupts the flow from the overarching structure of the paper laid out in L53-70 and Sec. 3 (i.e., "sorbent role"). Perhaps this material could be shortened and included in the intro section before laying out the paper objectives/structure**

We agree with this helpful comment. Material has been shortened and added to the introduction of this manuscript. This section reads now as: Starting at the line 40: Since C and N cycles are interconnected in soils (Feng et al., 2019; Gärdenäs et al., 2011), they should be regulated by the same factors, including mineralogy type (Wade et al., 2018). Increasing evidence shows that Fe specifically represents a major control over N biological transformations, including mineralization (Wade et al., 2018), nitrification (Huang et al., 2016a) (Han et al., 2018) denitrification (Zhu et al., 2013) (Wang et al., 2016), as well as their abiotic analogous reactions, such as chemo-denitrification (Burger and Venterea, 2011) and Fe-mediated hydroxylamine oxidation to nitrous oxide ($N_2O$). These reactions and others Fig.1, are likely to operate ubiquitously in soils, due to the close proximity between Fe minerals and SOM since most of the latter is contained in association with the former (Lalonde et al., 2012; Wagai and Mayer, 2007). The characteristic properties of individual Fe minerals and N compounds and how these properties are influenced by the soil environment likely drive the aforementioned reactions as well. First, Fe exists in a variety of polymorphs (Navrotsky et al., 2008) and is a redox-sensitive element that cycles between Fe(II) and Fe(III) states as controlled by soil Eh and pH. While Fe(III) promotes N stabilization within mineral associations, Fe(III) mobilization when it is reduced to Fe(II) can release N into solution. Fe reactivity is also driven by the amount and sign of surface charge, surface topography, particle size, crystallinity (Li et al., 2015a) (Petridis

et al., 2014) and the presence and the type of organic matter (OM) coverage (Gao et al., 2018; Kleber et al., 2007; Poggenburg et al., 2018) (Boland et al., 2014; Henneberry et al., 2016) (Kaiser and Zech, 2000a) (Daugherty et al., 2017). Second, soil N exists predominantly in organic forms (ON); mostly as protein and peptides, and to a lesser extent as amino-sugars and nucleic acids (Kögel-Knabner, 2006) (Knicker, 2011; Schulten and Schnitzer, 1997). Proteins are intrinsically reactive towards soil minerals, due to a number of properties, including hydrophobicity, surface charge distribution, surface area, number and type of functional groups, conformation, and size (Lützow et al., 2006). N from these compounds is generally not directly bioavailable due to molecular size constraints on microbial cell uptake (Schimel and Bennett, 2004). Depolymerization reactions, carried out by the activity of extracellular enzymes, such as peptidases, transform these polymers into soluble, low molecular weight (MW) organic monomers (e.g., short oligopeptides, amino acids (AAs)). Recent research shows that the size of AAs available for mineralization is controlled by peptidase activity, but more so by protein availability, both of which are affected by the interactions with Fe minerals. Therefore, Fe may drive gross AA production in soils (Noll et al., 2019).

5. **L127. This question doesn't lead well into the overarching theme of this section and detracts from the main thread. Start with an introductory paragraph that introduces the topic and provides a roadmap for the rest of the section. I assume that the "sorbent role" isn't just about enzymes, right?**

 **Response:** We agree with this helpful comment. An introductory paragraph has been added to all the sections.

6. **Sec. 4. As noted above, starting the section with a question is not the best way to introduce the topic and provide an overview for the section. In the first paragraph, lay out what the questions or topics are and then follow up with a succinct discussion of each. Ditto this comment for all sections.**

 **Response:** We agree with this helpful comment. An introductory paragraph has been added to all the sections. For section 4, this paragraph reads as: "The impact of structural Fe on N bioavailability in soils is a complex phenomenon that can be influenced by various factors. This section provides a detailed examination of this subject, focusing on how the structural Fe in clay and aggregates influence N turnover, as well as Fe- induced organic nitrogen (ON) polymerization"

7. **Fig. 2. I would encourage you to think about how this figure could be a bit more nuanced rather than just having "clouds" for enzymes, N substrates, etc.**

**Response:** This figure has been eliminated from the manuscript to address the first reviewer comment. An updated figure has been incorporated into the manuscript at line 290.

8. **Sec. 5. You skip the "electron transfer role" and go straight to "catalytic role". If there isn't a separate section for "electron transfer role" then remove from overview paragraph as a separate, defined "role".**

**Response:** The electron Transfer role of Fe is presented in lines 330-355

9. **Sec. 7 and 8. As noted by the other reviewer, these sections lack structure and were not adequately introduced earlier in the manuscript.**

Since we rewrote the whole document, the numbering and subheadings have been reimagined to accommodate new changes. This is the overall structure of the document:

1. Introduction
2. Roles of Fe in controlling N bioavailability
3. Involvement of Fe in soil phenomena that affect N bioavailability
4. Impact of global change on Fe-N bioavailability interactions
5. Synthesis and outlook

Each section has an introductory paragraph to guide the reader through the information provided within that section.

Here is an example at the line 450:

**4 Impact of global change on Fe-N bioavailability interactions**

Anticipated future climate scenarios indicate substantial fluctuations in precipitation and temperature patterns, accompanied 450 by increasing levels of atmospheric $CO2$. These changes, along with alterations in land use, have the potential to significantly impact Fe-N bioavailability interactions in various ways, which are detailed below.

All the figures of the manuscript have been updated with colors and revised. We also introduced information from all the relevant review papers proposed by the reviewer to build a case for the role of Fe

in N bioavailability (line 38-40). That was very helpful Thank you !! We hope these changes are satisfactory and address the reviewer's concerns.

**Responses to the last reviewer comments**

**Comment: fungi do not necessarily have low CUE compared to bacteria.**

We thank the reviewer for their comments on this version of the manuscript. The manuscript was checked for all the helpful suggestions and corrections that were minor. We just shed light that we changed the information in line 400 to "Beyond SOM stabilization, Fe oxides may regulate priming by altering microbial community composition, soil C and N content (Heckman et al., 2009; Heckman et al., 2018); potentially by restricting nutrient availability and changing the structural properties of dissolved organic matter (DOM). For instance, the application of goethite to soil limits P and N bioavailability while increasing the aromatic content of water extractable organic matter (WEOM), which may lower the ratio of fungi to 400 bacteria (Heckman et al., 2012) and alter C and N cycling as a consequence (Silva-Sánchez et al., 2019; Wardle et al., 2004)."

We hope this version of the manuscript is successful.